# Minimally Invasive Management and Diagnosis of Ectopic Splenic Pregnancy

**DOI:** 10.3390/medicina60030470

**Published:** 2024-03-13

**Authors:** Vestina Strakšytė, Rūta Mikšytė, Ugnė Kulnickaitė, Simas Gudelevičius

**Affiliations:** 1Department of Radiology, Lithuanian University of Health Sciences, 50009 Kaunas, Lithuania; ugne.kulnickaite@gmail.com; 2Medical Academy, Lithuanian University of Health Sciences, 44307 Kaunas, Lithuania; ruta.miksyte@stud.lsmu.lt (R.M.);

**Keywords:** ectopic splenic pregnancy, computed tomography, active extravasation, coil embolization

## Abstract

This report describes the case of a 32-year-old woman with ectopic pregnancy in the spleen, which was complicated by active bleeding. The patient complained of intermittent pain in her left side and lower abdomen that lasted several days. The serum beta-human chorionic gonadotropin (β-hCG) was increased, but no intrauterine gestational sac was found via transvaginal sonography. A computed tomography (CT) examination revealed the presence of a heterogeneous structure in the left peritoneal cavity, inferior to the spleen; signs of active extravasation; and a large amount of hemorrhagic fluid in the pelvis. An angiography examination also showed slow active extravasation from a small artery that branches off at the lower pole of the spleen. Coil embolization was performed. Splenic ectopic pregnancy can be managed by minimally invasive methods in carefully selected patients.

## 1. Introduction

Ectopic pregnancy occurs during the development of the blastocyst embryo upon implantation anywhere but the uterine cavity [1]. The most common site of ectopic implantation is within a fallopian tube, with a frequency of occurrence of 98% [1]. Clinicians tend to neglect the abdominal cavity and focus more on the diagnosis of tubal pregnancies.

A pregnancy that develops in the abdominal cavity instead of the uterus is called an abdominal ectopic pregnancy [2]. It occurs in approximately 1.2–1.4% of all reported pregnancies worldwide [3]. Splenic pregnancy is one of the least common forms of abdominal ectopic pregnancy, with scarce data [4].

The frequency of ectopic pregnancy is increasing due to the growing numbers of cesarean sections, pelvic inflammatory disease, and sexually transmitted diseases, as well as the use of artificial reproduction techniques [5,6]. 

The implantation site of ectopic pregnancy affects the severity of complications. With advances in its early diagnosis and management, mortality caused by ectopic pregnancy has decreased [7]. Nevertheless, abdominal ectopic pregnancies have a significant morbidity and mortality rate due to the risk of bleeding from the placental implantation site. Once it ruptures, it can lead to uncontrollable bleeding and other complications such as shock, or even death. In order to lower morbidity and death rates, it is crucial to effectively recognize and treat this uncommon kind of pregnancy [8,9]. 

The current standard for the diagnosis of ectopic pregnancy includes transvaginal (TVUS) or transabdominal ultrasound (TAUS) and the monitoring of β-human gonadotropin (β-HCG) levels [10].

There is no single effective treatment for ectopic pregnancy. Appropriate treatment depends on the location of the ectopic pregnancy, as well as the patient’s condition. Therapeutic methods of this pathology include pharmacotherapy and various surgical techniques [6].

We report a case of splenic ectopic pregnancy complicated by hemoperitoneum, which was successfully managed conservatively. 

## 2. Case

A 32-year-old woman was admitted to the Emergency Department with weakness; strengthening and intermittent, severe pain in her left side and lower abdomen, lasting several days; nausea; and vomiting. The patient had a history of two previous ectopic pregnancies following right and left tubectomies. The latter was removed several months ago, accompanied by profuse intraoperative bleeding. The patient was expecting her menstrual period. Abdominal examination revealed significant findings of diffuse abdominal and rebound tenderness. The woman was hemodynamically stable.

A complete blood count test performed out of whole blood (Auto Hematology Analyzer BC-6800, Mindray, Shenzhen, China) showed leukocytosis (17.36 × 10^9^/L), thrombocytosis (505 × 10^9^/L), and a low hemoglobin concentration of 86 g/L. A urine pregnancy test was positive (Servoprax GmbH, Wesel, Germany, detection at 20 mlU/mL HCG). The serum β-hCG level was elevated to 3400 IU/L (Randox, Crumlin, UK, Immunoassay method, normal value range 0–5 IU/L). A vaginal examination showed slight vaginal bleeding. Transvaginal sonography (GE Voluson E8, vaginal probe IC 5-9) revealed a normal-sized uterus and no identifiable intrauterine gestational sac.

A computed tomography (CT) examination of the abdomen and pelvis was performed due to an unclear diagnosis, using a LightSpeed VCT 64 scanner (GE Healthcare, Chicago, IL, USA). The CT protocol consisted of a non-contrast phase followed by an arterial phase, acquired with an automatic bolus tracking technique, with the region of interest placed in the proximal abdominal aorta, as well as a venous phase. A bolus of 80 mL of Ultravist 370 (Schering AG, Berlin, Germany) followed by a 20 mL saline flush was administered using an automated contrast injection system at a flow rate of 3.5 mL/s. The CT images were reformatted in both the coronal and sagittal planes. An abdomen and pelvis CT showed a heterogeneous structure in the left peritoneal cavity, inferior to the spleen (Figure 1a,b). Signs of active extravasation, streaks of unevenly distributed fluid around the hematoma, and a large amount of hemorrhagic fluid in the pelvis were visible. The gestational sac with an embryo due to a large hematoma and signs of active extravasation was not visible, which impeded prompt diagnosis.

The final diagnosis of a splenic ectopic pregnancy was made based on a previous history of ectopic pregnancies, elevated β-hCG levels, a positive pregnancy test, TVUS that did not show an intrauterine pregnancy, and CT examination findings of a mass with signs of active bleeding. The patient’s condition required an urgent treatment decision.

The minimally invasive approach is the method of choice for treating abdominal active extravasation in our institution. Selective angiography was performed and showed slow active extravasation from a small artery that branches off the lower pole of the spleen (Figure 2). Coil embolization was performed to stop the bleeding (Figure 3).

Due to the high probability of heavy bleeding and the possible complexity of the surgery, the patient was treated conservatively, and a two-dose regimen of methotrexate was prescribed on the first and fourth days. On the following day, the β-hCG level lowered to 2900 IU/L. The patient was followed up with an abdomen ultrasound, with monitoring of the dynamics of the hematoma (Figure 4).

Serum β-hCG levels decreased to 99.1 IU/L after 20 days, after which the patient was discharged from the hospital. Due to the COVID-19 pandemic, a decision was made to monitor the dynamics of β-hCG and the size of the hematoma near the spleen by abdominal ultrasound in the hospital. After two months, the β-hCG level normalized to 1.8 IU/L. The abdomen CT examination was repeated after seven months (Figure 5). The mass near the spleen had decreased in size. The pain under the left rib cage had disappeared. The patient’s recovery was uneventful. 

## 3. Discussion

Splenic ectopic pregnancy is an extremely rare type of ectopic pregnancy and carries a high risk of life-threatening intraperitoneal bleeding [11]. As with the liver, the spleen is relatively favorable for implantation because it is a flat and vascular organ that is easily accessible by the fertilized ovum in the supine position [12]. However, neither the spleen nor the liver can accommodate placental attachment or a growing embryo and they carry a high risk of rupture and massive intraperitoneal hemorrhage [13]. 

Risk factors associated with abdominal pregnancy align with those of other ectopic pregnancies and include a history of previous ectopic pregnancies, pelvic inflammatory disease, the presence of an intrauterine device (IUD), endometriosis, in vitro fertilization and prior tubal surgeries, smoking, and infertility [10,11,13]. Patients with a history of one prior ectopic pregnancy have a 10% risk of subsequent ectopic pregnancy recurrence. In contrast, it is greater than 25% in those with a history of two or more prior pregnancies [10]. In our case, the patient had undergone bilateral tubectomies due to two previous ectopic pregnancies, and this is her third ectopic pregnancy. Thus, knowing the patient’s history well, we can expect a high risk of ectopic pregnancy. There is no scientific evidence in the literature that supports this method of the prevention of repeated ectopic pregnancy.

The nature, location, and intensity of pain in ectopic pregnancy can vary. It often begins as colic pain in the abdomen or pelvis, localized to one side as the pregnancy expands the fallopian tube. The pain may become more generalized when the tube ruptures and the hemoperitoneum develops. Other possible symptoms include diarrhea or pain during bowel movements, vomiting, presyncope, syncope, shoulder pain, lower urinary tract symptoms, rectal pressure, and vaginal bleeding [13,14].

The clinical presentation of splenic ectopic pregnancy is associated with many symptoms and can lead to misdiagnosis. Splenic pregnancies present earlier than other abdominal extrauterine pregnancies, mainly manifesting as an acute abdomen and hemoperitoneum occurring at 6–8 weeks after gestation [13]. Therefore, in any female of reproductive age presenting with atypical abdominal pain, the possibility of an ectopic abdominal pregnancy must be considered [13].

β-hCG during pregnancy can be detected as early as eight days after ovulation [13]. Although the β-hCG level is 1500 IU/L or higher, ectopic pregnancy should be suspected in patients whose intrauterine gestational sac cannot be seen on TVUS. In the diagnosis of extrauterine pregnancy, both β-hCG measurement and TVUS are necessary. There is no consensus in the literature regarding the cutoff level of β-hCG for expectant treatment that results in the highest success rate. It ranges from 200 to 1500 IU/L [5].

Patients with β-hCG levels above the discriminatory zone and no intrauterine pregnancy identified on ultrasound require additional evaluation to determine the location of the ectopic pregnancy [15]. 

Abdominal ultrasound is the primary and vital imaging modality in all cases. A significant advantage of US is accessibility within the consulting room, minimizing the need for the patient to move and reducing the risk of gestational sac rupture [16]. The ultrasound diagnosis of nontubal ectopic pregnancy can be challenging and requires experienced sonographers who can identify atypical and rare locations of a gestational sac [1].

In situations of splenic or other abdominal pregnancy suspicion, an abdominal CT scan or MRI should be performed to evaluate the entire abdominal cavity [16]. CT and MRI scans are the most reliable in describing the relationship between the location of the gestational sac and surrounding tissue and in assessing the risk of rupture and other complications [16,17]. In our case, a CT scan was performed because the patient complained of diffuse abdominal pain, hemoglobin was low (86 g/L), and no pregnancy was observed in the uterus.

Patient preferences, clinical findings, ultrasound findings, and β-hCG levels should inform the decision to treat the extrauterine pregnancy medically or surgically [14].

There is limited evidence for the management of nontubal ectopic pregnancies. There is no consensus on the criteria for selecting a patient for medical versus surgical treatment or the optimal dosing protocols for medical treatment. The types of operations performed depend on the site, local expertise, resources, and patient characteristics [1]. 

Several methotrexate regimens have been studied, including single-, two-, and multi-dose protocols. The two-dose protocol is more effective than the single-dose protocol in patients with higher baseline β-hCG levels, but the latter carries the lowest risk of adverse effects [14]. Moreover, Po et al. offer a systematic multi-dose protocol for abdominal pregnancy [1]. 

Surgical treatment is the traditional treatment for traumatic splenic hemorrhage. However, severe infection may occur after a splenectomy since the spleen is essential to the human body’s immune system [18]. 

Partial or total splenectomy is the most common treatment in the literature [13,19]. Laparoscopic spleen surgery is a minimally invasive and effective method. However, laparotomy may be the best choice when the patient is hemodynamically unstable and presents with a massive hemoperitoneum [13].

In fact, with recent advances in interventional radiology and diagnostic CT, nonoperative management, including transcatheter arterial embolization (TAE), has become the treatment of choice. TAE is required if the CT findings suggest extravasation, pseudoaneurysm, or vessel disruption [20].

In recent cases, there have been successful reports of conservative treatment using systemic or local methotrexate or a combination of methotrexate and potassium chloride injections [21,22]. To our knowledge, only one case of splenic pregnancy treated with selective embolization of the feeding vessels has been reported in the English literature. In this case, the patient was stable without any signs of active extravasation [23]. We presented the first case of splenic pregnancy when it was emergent and demanded urgent solutions. Minimally invasive treatments offer an advantage by preserving the patient’s spleen and reducing the risk of immunosuppression and infection [24].

## 4. Conclusions

In this paper, we present the first case of splenic ectopic pregnancy that was complicated with active bleeding (hemoperitoneum). In this case, a conservative approach, including angiography coil embolization and a two-dose regimen of methotrexate, successfully averted the need for more invasive surgical intervention. Our successful experience demonstrates that splenic preservation should be considered when possible. 

## Figures and Tables

**Figure 1 medicina-60-00470-f001:**
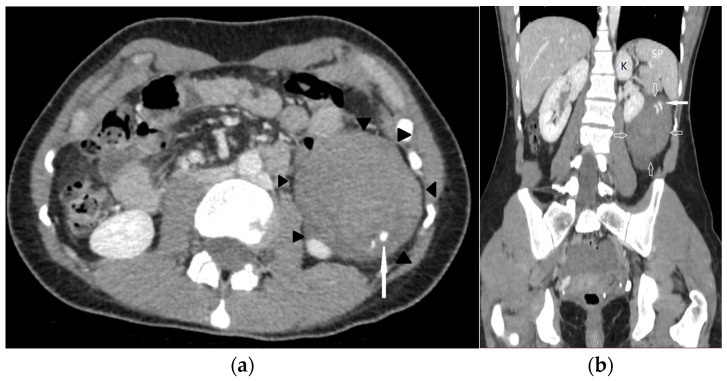
The contrast-enhanced abdomen–pelvis CT images: (**a**) (axial) and (**b**) (coronal) images revealed a heterogenous mass of 7.5 × 6.3 cm (black arrowheads) with signs of active extravasation (arrow), which was inferior to the spleen (SP) and kidney (K).

**Figure 2 medicina-60-00470-f002:**
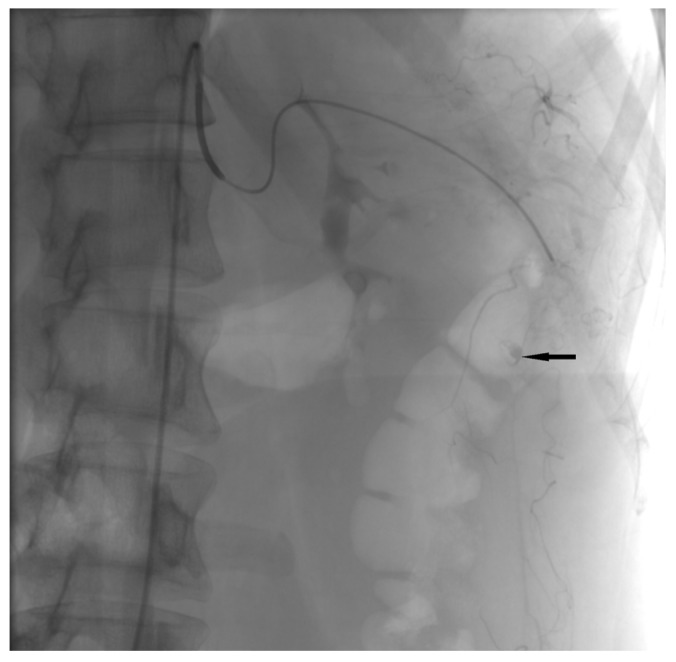
The angiogram showed active extravasation (arrow).

**Figure 3 medicina-60-00470-f003:**
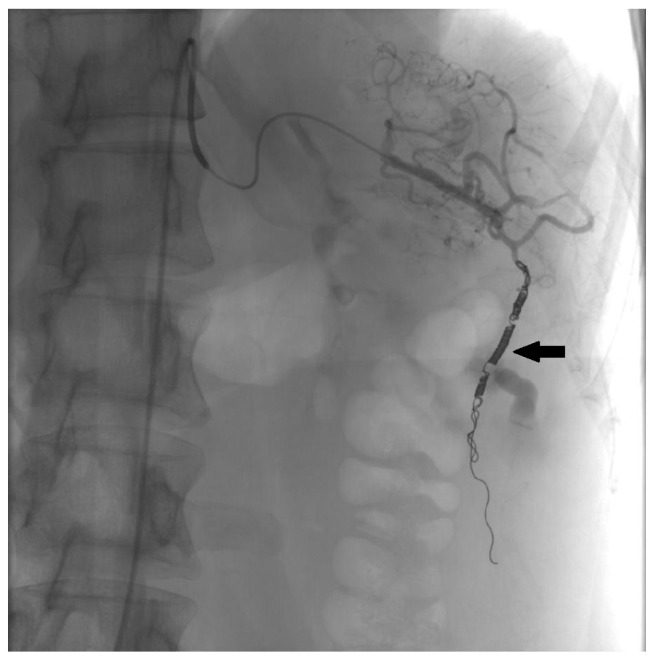
The post-embolization angiogram revealed platinum coils (arrow) implanted in the minor splenic artery branch.

**Figure 4 medicina-60-00470-f004:**
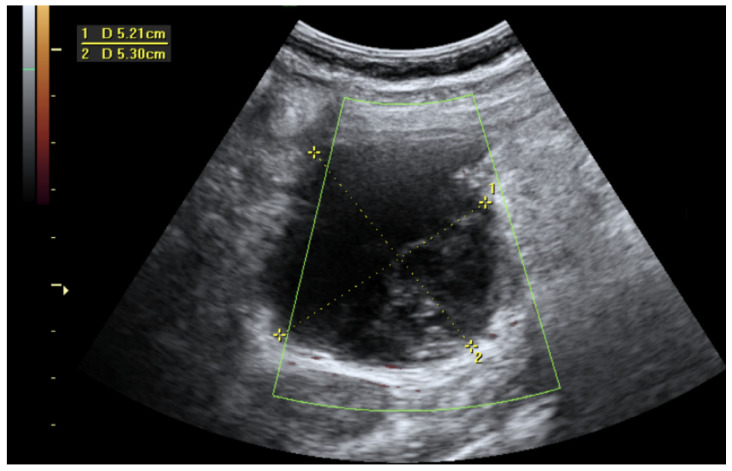
An ultrasound image was performed one month later. The hematoma was reduced in size and was liquefying.

**Figure 5 medicina-60-00470-f005:**
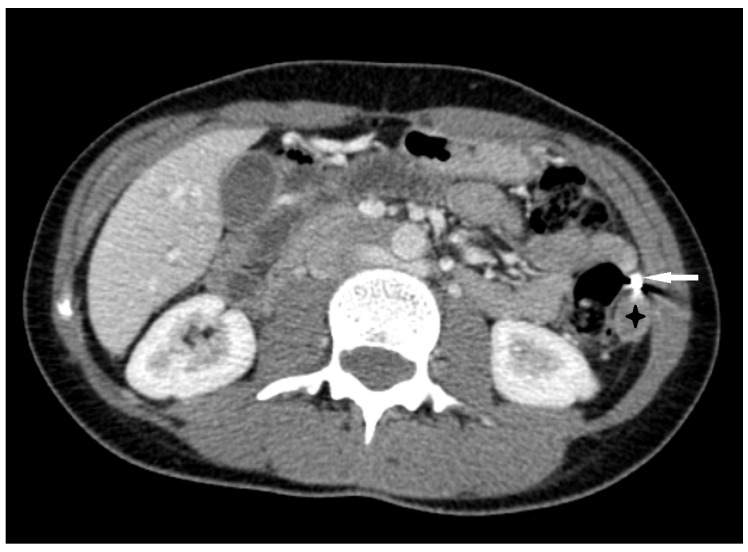
A CT scan was obtained seven months after coil embolization and conservative treatment. There were significant changes in heterogenous mass size. It decreased significantly (asterisk). Note the artifacts from the inserted coils (arrow).

## Data Availability

The data and materials used in this study are available upon reasonable request from the corresponding author. Restrictions may apply to the availability of specific data sets due to privacy or ethical restrictions.

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
