# Peer review of "Minimally Invasive Management and Diagnosis of Ectopic Splenic Pregnancy"

_medicina, 2024, doi:10.3390/medicina60030470_

Round 1

Reviewer 1 Report

Comments and Suggestions for Authors

Journal

Medicina (ISSN 1648-9144)

Manuscript ID

medicina-2894461

Type

Case Report

Title

Minimally Invasive Management of Ectopic Splenic Pregnancy

Authors

Vestina Strakšytė * , Rūta Mikšytė , Ugnė Kulnickaitė , Simas Gudelevičius

Section

Surgery

Special Issue

New Insights in Bleeding: From Diagnosis to Therapy

Abstract

The publication is an unique case report. The authors presented a case of a 32-year-old woman with an ectopic pregnancy in the spleen.

The woman complained of atypical intermittent pain in the left side and lower abdomen for long time

The serum beta-human chorionic gonadotropin (β-HCG) was eelvated, but they coud not confirm intrauterine gestation by ultrasound. Examination by computed tomography (CT) revealed the presence of a heterogeneous structure in the left peritoneal cavity, inferior to the spleen, signs of active extravasation, and a large amount of hemorrhagic fluid in the pelvis. Angiography examination also showed slow active extravasation from a small artery that branches off at the lower pole of the spleen. The cure was embolization. The splenic ectopic pregnancy can be managed by minimally invasive method.

Case description

How was determined the WBC, PLT, Hb levels? Give the machine and method description. Did you apply plazma or whole blood for this test?

What kind of test confirm the gestation? Urinary test? Which company distributes the test? Give the sensitivity level for beta HCG

Give laboratory parameters of Serum beta HCG measurement (company, method, ID, etc)

Give transvaginal ultrasound description (machine name, trasnducer parameters)

Give CT investigation dexription (machine name, technikque)

How could you confirm the fact, that the a heterogenous mass with signs of active extravasation in splen was gestation? The ultrasound and CT images did not present embrio, yolk sec, gestation sac.

Author Response

Dear Reviewers:

We want to thank you and the referees for reviewing our manuscript entitled “ Minimally Invasive Management and Diagnosis of Ectopic Splenic Pregnancy”.Thank you all for the very helpful suggestions and comments, which have encouraged us to improve the manuscript. We have carefully addressed the issues raised by the referees. Our detailed responses to the comments and questions of the referees are listed below.

Vestina Strakšytė

Rev 1

How was determined the WBC, PLT, Hb levels? Give the machine and method description. Did you apply plazma or whole blood for this test?

Our response: A complete blood count test performed out of whole blood (Auto Hematology Analyzer BC-6800, Mindray).

What kind of test confirm the gestation? Urinary test? Which company distributes the test? Give the sensitivity level for beta HCG

Our response: A urine pregnancy test was positive (Servoprax GmbH, detection at 20 mlU/ml HCG).

Give laboratory parameters of Serum beta HCG measurement (company, method, ID, etc)

Our response: The serum β-HCG was performed by the Randox Immunoassay method, with a normal value range 0-5 IU/L.

Give transvaginal ultrasound description (machine name, trasnducer parameters)

Our response: Transvaginal sonography (GE Voluson E8, vaginal probe IC 5-9)

Give CT investigation dexription (machine name, technikque)

Our response: The (CT) examination was performed using a LightSpeed VCT 64 scanner (GE Healthcare, USA). The CT protocol consisted of a non-contrast phase followed by an arterial phase, acquired with an automatic bolus tracking technique with the region of interest placed in the proximal abdominal aorta and a venous phase. A bolus of 80 mL of Ultravist 370 (Schering AG, Berlin, Germany) followed by a 20 mL saline flush was administrated using an automated contrast injection system at a flow rate of 3.5 mL/s. The CT images were reformatted in both coronal and sagittal planes.

How could you confirm the fact, that the a heterogenous mass with signs of active extravasation in splen was gestation? The ultrasound and CT images did not present embrio, yolk sec, gestation sac.

Our response: The gestational sac with embryo due to a large hematoma and signs of active extravasation was not visible, which impeded prompt diagnosis.

The final diagnosis of splenic ectopic pregnancy was made based on a previous history of ectopic pregnancies, elevated β-HCG, positive pregnancy test, the TVUS that did not show an intrauterine pregnancy, and the CT examination findings of mass with signs of active bleeding. The patient's condition required an urgent treatment decision.

Reviewer 2 Report

Comments and Suggestions for Authors

Review Minimally Invasive Management of Ectopic Splenic Pregnancy

Thank you for allowing me to review this manuscripts, it is very interesting, showing a successful and original management of a very rare type of ectopic pregnancy. I believe all clinicians would find this interesting. I believe mentioning the radiological approach in the title could offer better insights to the readers, and radiologist could also be interested in the paper.

Nevertheless, some issues need clarification and more details are needed.

Lines - 33-34 – the readers could understand that the ectopic pregnancies occurred AFTER the salpingectomies, not that they were treated by salpingectomy? 

Taking this recurrence into consideration, what prophylactic measures were implemented after the first two diagnoses? Also, what possible measures could be taken into account in this case in order 

to prevent another recurrence?

Congratulations for choosing the CT investigation that actually lead to a successful diagnosis and treatment! Also, for choosing to do an abdominal, not only a pelvic examination (considering that tubal localization of ectopic pregnancy is overwhelmingly frequent, clinicians would usually ask only for a pelvic exam). Please offer some details regarding the motives that lead to choosing this option, as it is not in the current protocols. 

Lines 46-47 – was the CT performed immediately after admission? The laboratory tests do not initially show anemia. What were the other clinical parameters? Were there clinical signs of hypovolemia? Could there be an approximation of the blood volume that was lost?

Also, as surgical intervention is the golden standard for ectopic pregnancy with hemodynamic instability/hemoperitoneum, please offer more details regarding this aspect – was a laparoscopy performed? If so, what did it reveal? If not, why was it not preformed, as it would also have been a life-saving procedure?

Line 69 – the patient was admitted in the hospital for over 20 days? This appears unusually long. Were there complications? What was her evolution, why did she need such a long hospital stay? Or was it just prophylactic? Please offer more details regarding the hospitalization.

Was there at any moment the possibility of a laparoscopic evacuation of the hematoma?

Lines 100-101 – reference needed, I personally do not agree with this affirmation

Line 108 – Hb was low – the statement needs to be backed up by laboratory findings, as this is not mentioned in the case description

The details mentioned in the discussion section are very general affirmations and offer little information to the clinicians. Maybe more details regarding the actual procedure, the potential risks, potential damage to the splenic parenchyma, possibility of collateral vessels etc. could bring more interesting scientific soundness to the paper.

Comments on the Quality of English Language

 Lines - 33-34 – the readers could understand that the ectopic pregnancies occurred AFTER the salpingectomies, not that they were treated by salpingectomy? 

Author Response

Dear Reviewers,

We want to thank you and the referees for reviewing our manuscript entitled “ Minimally Invasive Management and Diagnosis of Ectopic Splenic Pregnancy”.Thank you all for the very helpful suggestions and comments, which have encouraged us to improve the manuscript. We have carefully addressed the issues raised by the referees. Our detailed responses to the comments and questions of the referees are listed below.

Vestina Strakšytė

Rev 2

Thank you for allowing me to review this manuscripts, it is very interesting, showing a successful and original management of a very rare type of ectopic pregnancy. I believe all clinicians would find this interesting. I believe mentioning the radiological approach in the title could offer better insights to the readers, and radiologist could also be interested in the paper.

Nevertheless, some issues need clarification and more details are needed.

Lines - 33-34 – the readers could understand that the ectopic pregnancies occurred AFTER the salpingectomies, not that they were treated by salpingectomy? 

Taking this recurrence into consideration, what prophylactic measures were implemented after the first two diagnoses? Also, what possible measures could be taken into account in this case in order 

to prevent another recurrence?

Our response: We have clarified and added more details.

A 32-year-old woman was admitted to the Emergency Department with weakness, strengthening, and intermittent severe pain in the left side and lower abdomen, lasting several days, nausea, and vomiting. The patient had a history of two previous ectopic pregnancies following right and left tubectomies. The left one was removed several months ago, accompanied by profuse intraoperative bleeding.

There are no data in the literature, and they are not supported by scientific evidence on how to prevent repeated ectopic pregnancy.

Congratulations for choosing the CT investigation that actually lead to a successful diagnosis and treatment! Also, for choosing to do an abdominal, not only a pelvic examination (considering that tubal localization of ectopic pregnancy is overwhelmingly frequent, clinicians would usually ask only for a pelvic exam). Please offer some details regarding the motives that lead to choosing this option, as it is not in the current protocols. 

Our response: Thank you for noticing. The CT was performed of the abdomen and pelvis. We have corrected the text.

The abdomen and pelvis computed tomography (CT) was performed due to an unclear diagnosis.

Lines 46-47 – was the CT performed immediately after admission? The laboratory tests do not initially show anemia. What were the other clinical parameters? Were there clinical signs of hypovolemia? Could there be an approximation of the blood volume that was lost?

Our response: CT was performed after the initial abdomen and transvaginal ultrasound. The blood test showed anemia. It was mentioned in the text as having „low hemoglobin concentration of 86 g/L“.

Also, as surgical intervention is the golden standard for ectopic pregnancy with hemodynamic instability/hemoperitoneum, please offer more details regarding this aspect – was a laparoscopy performed? If so, what did it reveal? If not, why was it not preformed, as it would also have been a life-saving procedure?

Our response: No, the laparoscopy wasn‘t performed because the patient felt better, and the hematoma and HCG levels decreased.

In fact, with recent advances in interventional radiology and diagnostic CT, nonoperative management, including transcatheter arterial embolization (TAE), has become the treatment of choice. TAE is required if the CT findings suggest extravasation, pseudoaneurysm, or vessel disruption.].

Line 69 – the patient was admitted in the hospital for over 20 days? This appears unusually long. Were there complications? What was her evolution, why did she need such a long hospital stay? Or was it just prophylactic? Please offer more details regarding the hospitalization.

Was there at any moment the possibility of a laparoscopic evacuation of the hematoma?

Our response:  We have clarified. Due to Covid- 19 pandemic, a decision was made to monitor the dynamics of β-HCG and the size of hematoma near the spleen by abdominal ultrasound in the hospital.

Lines 100-101 – reference needed, I personally do not agree with this affirmation

Our response: Thank you. We have clarified. The nature, location, and intensity of pain in ectopic pregnancy vary. It often begins as a colic pain in the abdomen or pelvis, localized to one side as the pregnancy expands the fallopian tube. The pain may become more generalized when the tube ruptures and the hemoperitoneum develops. Other possible symptoms include diarrhea or pain during bowel movements, vomiting, presyncope, syncope, shoulder pain, lower urinary tract symptoms, rectal pressure, and vaginal bleeding [13, 14].

 Line 108 – Hb was low – the statement needs to be backed up by laboratory findings, as this is not mentioned in the case description

Our response: We have clarified In our case, a CT scan was performed because the patient complained of diffuse abdominal pain, hemoglobin was low (86 g/L), and no pregnancy was observed in the uterus.

The details mentioned in the discussion section are very general affirmations and offer little information to the clinicians. Maybe more details regarding the actual procedure, the potential risks, potential damage to the splenic parenchyma, possibility of collateral vessels etc. could bring more interesting scientific soundness to the paper.

Our response: We have improved this section and added more details about risk factors, management and etc.

Round 2

Reviewer 2 Report

Comments and Suggestions for Authors

I appreciate most of the addings and changes in the manuscript and believe it is better suited for publication.

Comments on the Quality of English Language

Minor language issues are still present in the manuscript, maybe a native speaker or other form of medical English checking could be performed.

Author Response

Dear Reviewer,

Thank you for your comment. The English language was improved. Please find the attached certificate.

Thank you.

Sincerely,

Vestina
